# Dry Powder for Pulmonary Delivery: A Comprehensive Review

**DOI:** 10.3390/pharmaceutics13010031

**Published:** 2020-12-28

**Authors:** Birendra Chaurasiya, You-Yang Zhao

**Affiliations:** 1Program for Lung and Vascular Biology, Stanley Manne Children’s Research Institute, Ann & Robert H. Lurie Children’s Hospital of Chicago, Chicago, IL 60611, USA; bchaurasiya@luriechildrens.org; 2Department of Pediatrics, Division of Critical Care, Northwestern University Feinberg School of Medicine, Chicago, IL 60611, USA; 3Department of Pharmacology, and Department of Medicine (Division of Pulmonary and Critical Care Division), Northwestern University Feinberg School of Medicine, Chicago, IL 60611, USA; 4Feinberg Cardiovascular and Renal Research Institute, Northwestern University Feinberg School of Medicine, Chicago, IL 60611, USA

**Keywords:** respiratory architecture, dry powder formulation, physicochemical properties, clinical factors, pulmonary medicines

## Abstract

The pulmonary route has long been used for drug administration for both local and systemic treatment. It possesses several advantages, which can be categorized into physiological, i.e., large surface area, thin epithelial membrane, highly vascularized, limited enzymatic activity, and patient convenience, i.e., non-invasive, self-administration over oral and systemic routes of drug administration. However, the formulation of dry powder for pulmonary delivery is often challenging due to restrictions on aerodynamic size and the lung’s lower tolerance capacity in comparison with an oral route of drug administration. Various physicochemical properties of dry powder play a major role in the aerosolization, deposition, and clearance along the respiratory tract. To prepare suitable particles with optimal physicochemical properties for inhalation, various manufacturing methods have been established. The most frequently used industrial methods are milling and spray-drying, while several other alternative methods such as spray-freeze-drying, supercritical fluid, non-wetting templates, inkjet-printing, thin-film freezing, and hot-melt extrusion methods are also utilized. The aim of this review is to provide an overview of the respiratory tract structure, particle deposition patterns, and possible drug-clearance mechanisms from the lungs. This review also includes the physicochemical properties of dry powder, various techniques used for the preparation of dry powders, and factors affecting the clinical efficacy, as well as various challenges that need to be addressed in the future.

## 1. Introduction

Anatomically, lungs are bifurcated into the right and left lung; the right lung is further divided into three lobes, while the left lung is divided into two lobes. The air passages continue to divide into smaller capillary tubes, from the larynx, trachea, and bronchi to tiny sacs, namely alveoli [1]. The pulmonary route of drug administration has gained popularity due to its several advantages over other drug administration routes, including (1) rapid drug absorption owing to the thin epithelial membrane lining in the lungs, high density of blood vessels, and the large surface area of lungs; (2) non-invasive drug administration; (3) minimal risk for enzymatic degradation of drugs as the lung has low enzymatic activity. Although the pulmonary route has been an accepted route of drug administration for both local and systemic treatments, formulation of pharmaceuticals for pulmonary delivery can prove challenging due to its limited tolerance for foreign particles [2,3,4,5,6]. Despite the many advantages of pulmonary delivery, some studies have reported that the route is hindered by some drawbacks, including local irritation in the tracheal tube and quick clearance from the site of deposition [7]. Removal of inhaled powders is believed to occur by mucociliary clearance, which involves phagocytosis via alveolar macrophages and the local dendritic cells present on the lining of the tract’s epithelial region [8]. This results in lower concentrations of delivered doses at the target site. Several approaches have been used to overcome these obstacles and improve the bioavailability of administered drugs. This includes manipulation of particle morphology (size and shape) for optimal inhalation, coating of the exterior surface of respirable particles with pulmonary-compatible materials [9], maximal deposition, and efficient escape from macrophage recognition [10]. While 1–5 µm is considered as the best size range for particle deposition in the alveolar region [11,12], it has also been reported that round particles or particles sized 1.5–3 µm are more susceptible to phagocytosis by alveolar macrophages [13].

Physiochemical properties of dry powders, such as size and shape, surface morphology and charge, hygroscopicity, and moisture content have direct effects on aerosolization, discharge from treatment devices, and bioavailability of aerosolized drugs. Dry powders designed for inhalation are very fine and can easily form agglomeration due to cohesion between individual particles and are hard to aerosolize [14]. It is estimated that dry powders with a mass median diameter of 1–5 µm and a bulk density less than 0.4 g/cm^3^ are suitable to form aerosol for deep lung deposition.

The most common technique for optimizing the physicochemical properties of dry powder to enhance inhalation is mechanical milling [15]. This technique is simple and economical but has various technical problems like inconsistent morphology, thermodynamically activated surfaces, and high electrostatic charges of the particles arising from high share impaction forces during production [16,17]. Other improved promising techniques reported are spray drying [18,19,20,21], spray-freeze drying [19,21,22,23,24], and supercritical fluid-carbon dioxide drying technique [15,25,26,27].

In this review, we have summarized the roles of the architecture of pulmonary airways and the physicochemical properties of dry powders in aerosolization, inhalation, deposition, and clearance. We also have described various techniques for the preparation of dry powder for inhalation, factors affecting the clinical efficacy, and various challenges related to dry powder preparation for inhalation.

## 2. Airways System

### 2.1. Respiratory Tract

Human pulmonary airway systems are divided into three regions, namely, the extra-thoracic region, the tracheobronchial region, and the alveolar region (Figure 1A) [28]. The extra-thoracic region includes the oral–pharyngeal cavity, larynx, and tracheal entrance, while the tracheobronchial region includes the trachea, bronchi, and bronchiole terminals. It is a complex system that can transport air from the trachea down to terminal bronchioles. This tracheobronchial system is partitioned into 23 generations of dichotomous branching, starting from the trachea (generation 0) to end up in terminal bronchioles (generation 23) [29]. The alveolar region is composed of bronchioles, alveoli, and alveolar ducts. There are approximately 300 million alveoli present in the lungs. The lungs are a highly vascularized organ in the human body, with more than 280 billion capillaries through which gaseous exchange takes place. The blood flow across the lung is as high as 5700 mL/min, which allows drugs administered through the pulmonary route to be absorbed rapidly with systemic effects [2]. Additionally, the architecture of the lung facilitates high levels of drug delivery when drugs are administered through the pulmonary route.

### 2.2. Particles Deposition Pattern

The extent of particle deposition in the respiratory tract is dependent on both the physiological conditions of the patient, including breathing patterns and the general health of the lungs, and physicochemical conditions of the inhaled particles, such as shape, size, bulk density, hygroscopicity, and moisture content [30,31]. After inhalation of particles, major mechanisms for deposition include impaction due to inertial forces, deposition due to gravity, and Brownian diffusion. Other mechanisms accounting for minor rates of deposition include interception and electrostatic precipitation [28]. Inertial impaction during inhalation exerts a centrifugal force on the aerosolized particles resulting in deposition throughout various bronchial regions depending on particle size. Larger particles (>5 µm) are deposited in the upper respiratory tract, whereas smaller particles (1–5 µm) get deposited in the bronchiolar region via sedimentation. Furthermore, particles < 1 µm are deposited in deeper alveolar regions through Brownian diffusion, while particles smaller than 0.5µm are exhaled out during exhalation [1,32]. This size-dependent particle sedimentation behavior in the respiratory tract has been well studied by Usmani et al. [33]. They prepared salbutamol aerosols of different particle sizes with aerodynamic diameters of 1.5, 3, and 6 µm and conducted a randomized, double-blind, placebo-controlled clinical experiment in healthy human volunteers. The study demonstrated that sedimentation of smaller-sized particles (1.5 and 3 µm) in the central and peripheral airways is much higher than that of larger-sized particles (6 µm). Furthermore, the study found that particles <1 µm have a high tendency to be expired out during breathing (Figure 1B).

### 2.3. Particle Clearance Mechanisms

The pulmonary system is divided into the conducting zone and the respiratory zone. The inhaled particulate matters have different mechanisms of clearance due to their solubility in different zones [34,35]. The main mechanism of clearance of insoluble particles from the conducting zone is the mucociliary escalator, in which ciliated epithelia move the insoluble particles towards the pharynx region. Foreign inhaled particles captured within the conducting zone take 15 min to 2 h to be cleared, with a speed of ~4–20 mm/min after inhalation [36]. Clearances of the deposited insoluble particles from the conducting zone also occur through phagocytosis by either alveolar macrophages moving up with the mucociliary escalator or macrophages entering the airways via bronchial and bronchiolar mucosa [37] and through epithelial endocytosis [38]. The majority of soluble particles from the conducting zone are cleared by an absorptive mechanism consisting of transepithelial permeation via intercellular pathways or by active and passive transcellular transport [39]. In addition, mechanical clearance along the mucociliary escalator or cough can contribute to tracheobronchial clearance of inhaled soluble particles. Chemical reactions can also influence the clearance rates of soluble substances from the conducting zone, i.e., reactions with and binding to cellular and extracellular components [40]. From the respiratory zone, insoluble particle matter is mostly cleared by alveolar macrophage phagocytosis mechanisms and transported towards the larynx by mucociliary escalation processes. Endocytosis by type I epithelial cells and subsequent exocytosis into the interstitium is another clearance mechanism from the deep lung [41,42]. Removal of free insoluble particles from respiratory bronchioles and alveolar ducts also occurs by mechanical mechanisms like dragging or fluid flux by continuous surfactant movement of the lung [40]. Similarly, transepithelial transport is the major mechanism for the clearance of inhaled soluble substances from the alveolar region. These soluble substances get diffused through the intercellular tight junction for clearance [43]. The clearance of particles from this region mainly depends on the nature of the particles, i.e., their molecular size and lipophilicity degree [35]. Other factors are lung volume, epithelial surface area, and distribution of the substance on the epithelial surfactant layer. Endocytosis by alveolar macrophages and type I epithelial cells can also contribute to solute clearance from the alveolar space of the lung [41]. Clearance of inhaled particulate matter from lungs based on their morphology and physical properties was extensively discussed in a review article by Liu et al. [35].

## 3. Physicochemical Properties of Dry Powder

The extent to which an aerosolized dry powder is delivered to the lungs is directly influenced by its physicochemical properties. Small-sized particles tend to clump together, forming agglomerates. Some of the physical properties of particles with major influence on aerosolization and release from the inhaler device include particle shape and size, hygroscopicity, moisture content, and electrostatic charge on the surface of particles.

### 3.1. Size of Particles

Typically, the aerosolization of dry powder for inhalation is based on its size and size distribution [45]. Size distribution can be calculated by span, which can directly affect the deposition of drugs in the lungs upon inhalation [46]. For effective aerosol formation to reach deep into the lung, a certain size distribution is required. To assess the quality of inhalational aerosol, various experimental techniques are used such as Anderson Cascade Impactor, Multistage Liquid Impinger and Next Generation Impactor, and the performance can be expressed by various quantitative parameters such as median diameter of the size distribution (Dv50), emitted dose (ED), fine particle fraction (FPF), fine particulate dose (FPD), mass median aerodynamic diameter (MMAD), and geometrical standard deviation (GSD) [47]. The value of Dv50 indicates that the particle size in micrometer is half of the total amount of dry powder delivered from the device during aerosolization. In other words, Dv50 divides the measured distribution into two halves, smaller and larger particle size. ED is the amount of drug exit from the device and is expressed in percentage. The total amount of dry powder in the range of 3–5 µm size can be calculated by interpolation from the inverse of the standard normal cumulative mass distribution minus stated cut-off size of respective stages of cascade impactor, while the smaller-sized particles (<3 µm) are calculated as FPF, expressed as a percentage of the ED or FPD. The MMAD value of particles can be measured using various cascade impactors, in which particles larger than a certain size that are not therapeutically relevant in inhalational medication are neglected [48]. This limiting size for Andersen Cascade Impactor is ~9 µm at the standard airflow rate of 28.3 L/min and ~8 µm for Next Generation Impactor at 60 L/min. Usually, the MMAD value of the particles is lower than the Dv50 value. Both values are equivalent only when the particle’ size falls below these limiting values. This MMAD value is a conceptual value and is central to any aerosol preparation for respiratory delivery. Theoretically, MMAD can be calculated from the geometric particle size and tap density [49]. GSD informs about the spread of particle size distribution around the mean value, i.e., Dv50 value, and can be measured in the form of percentile across the mean value of the 84th and 16th percentile [50].

Larger span value indicates higher heterogeneity in size distribution. Studies have shown that particles with diameters of 1–5 µm are best for preparing aerosols intended for inhalation [51]. Cohesiveness among particles is higher when the particles are smaller than 1 µm, whereas cohesiveness among particles is lower when the particles exceed 5 µm in size. Both high and low cohesiveness among particles is problematic for aerosol formation, while research suggests that particles in the range of 1–5 µm have relatively ideal levels of cohesiveness for optimal aerosolization.

Chew et al. have shown a relation among particle size, cohesiveness, and air shear force [14]. Furthermore, to demonstrate the influence of particle size and its distribution, the same group prepared three different-sized powders with aerodynamic sizes of 2.3, 3.7, and 5.2 µm of mannitol and used two different inhaler devices, i.e., Rotahaler^®^ and Dinkihaler^®^, to produce aerosol at 60 L/min and 120 L/min airflow [52]. They compared the relationship between size, aerosol formation, and resistance in both devices and found that the particles with aerodynamic sizes 2.7 µm and 5 µm aerosolized to 12% and 22% by weight, respectively, at 60 L/min supply of air. For Rotahaler^®^, under an air force of 120 L/min, the aerosolization of particles with aerodynamic size 2.7 µm increased to 25% by weight, while particles with 5 µm size were not significantly affected. For Dinkihaler^®^, 63% and 32% (*w*/*w*) of aerosolization were observed for particles with aerodynamic sizes of 2.7 µm and 5 µm, respectively, at the air force of 60 L/min. Interestingly, unlike what was observed with Rotahaler^®^, increasing the air force to 120 L/min reduced the aerosolization of particles for both sizes (Figure 2I).

### 3.2. Shape and Surface Morphology of Particles

Particle shape and surface morphology are the second most important factor that affects particle aerosolization and lung deposition [53] and have been the topic of various studies [53,54,55,56,57,58,59]. Particles with irregular shape have low contact area with low Van Der Walls forces, while additionally having a low tendency to form aggregates [60]. Hassan and Lau prepared particles of different shapes, for example, pollen-shaped, spherical, plate-shaped, cube-shaped, and needle-shaped (Figure 2II), with different techniques and studied the flowability, aerosolization, and deposition patterns of these particles (Figure 2III) [53]. Although the aerodynamic diameter of spherical, pollen, and cube-shaped particles were larger than plate- and needle-shaped particles, the FPF of spherical and pollen-shaped particles were higher. However, the cube-shaped particles were found to have lower FPF values compared to those of the needle- and plate-shaped particles. The surface morphology of pollen-shaped particles is rough and porous in comparison to spherical-, plate- and cube-shaped particles, which contribute to the pollen-shaped particles’ lower particle density. The aerodynamic diameters of these particles are smaller than their physical size. Additionally, the irregular surface of these particles prevents close contact between each other, resulting in reduced cohesion force for subsequent dispersion [61,62]. Chew and Chan have prepared two types of solid particles with similar size distribution (volume median diameter 3 µm, span 1.5 µm) but with different morphology using bovine serum albumin. One had a smooth surface (2.8 µm), and the other had a wrinkled (3.1 µm) surface with the same bulk density (1.2 g/cm^3^) [14]. They found that the wrinkled surface particles dispersed better than particles with smooth surfaces (Figure 2IV).

### 3.3. Hygroscopicity and Moisture Content

Hygroscopicity is the ability of a solid substance to absorb moisture from the surrounding environment. Solid materials continue absorbing moisture from the surroundings until equilibrium is reached with the surrounding environment. The moisture uptake by solid mass depends on their surrounding environmental conditions as well as the nature of the solid materials (i.e., lipophilic or hydrophilic) [63]. This phenomenon of gaining moisture from the environment affects many aspects of particles. For example, it increases bulk density of particles and alters surface charge and aerodynamic size of the powder [64]. Zhou et al. mentioned the hygroscopic property of spray-dried colistin powders, which significantly absorbed moisture up to 30% and had its FPF substantially reduced from 80% to 63.2% when stored at 60% humidity condition [65]. Furthermore, the powders stored at 90% humidity condition were found to clump together more often and were unable to aerosolize. This is mostly contributed to the high moisture environment’s effects on the inhalants. Emery et al. prepared hydroxypropyl methylcellulose (HPMC) and respitose powders with moisture contents of 0%, 2%, 5%, 10% and 0%, 2%, 5%, 8%, respectively, and found that aerosolization of HPMC gradually decreased with increased moisture content, whereas the aerosolization of respitose remained stable [66].

### 3.4. Surface Electrostatic Charge

The development of electrostatic charges on the surface of particles results from a number of factors that directly influence the aerosolization of the particles [67]. Surface charges on particles depend on their size and surface properties such as crystal lattice structures, surface energy, and surface area [68]. Large particles tend to have rough surfaces and irregular shapes compared to small particles [69], contributing to disorder within the crystal lattice and minor moisture uptake. Kaialy et al. evaluated the relationship between size and the surface charge of spray-dried mannitol. They found that the net electrostatic charge on the surface of mannitol particles increased from −0.1 ± 0.1 nanocoulomb/gram to 2.3 ± 1.4 nanocoulomb/gram as the mean size of the particles decreased from 122 µm to 45 µm (Figure 2V) [70]. Based on this correlation of size and net surface charge, it was concluded that smaller-sized particles provide more active surface area to transfer surface charge. The increased surface charge strengthens the cohesiveness among particles and between particles and the surface wall of the inhaler device and decreases the FPF. Similarly, the shape and surface morphology of particles also play a significant role in acquiring the surface charges [71]. Spherical-shaped particles are less prone to acquire charge in comparison to elongated particles [72], and particles with rough surfaces have high tendencies to exchange charges because of increased inter-particle and particle–surface contact areas [73,74]. Electrostatic charges on particles influence aerosolization during inhalation of dry powder as well. During inhalation, the powder aerosolizes in the device and gains large amounts of charge, which are further transmitted to the drugs [75]. Matsusyama and Yamamoto have shown that deposition of particles in the airways is also affected by surface charge [76]. Therefore, it is important to optimize the surface charge during the formulation of drugs [77]. The deposition pattern of charged particles in airways has been well explained using computational lung models [78,79]. Generally, deposition patterns of dry powder in the airways are governed by inertial impaction, gravitational sedimentation, and Brownian diffusion. However, electrostatic charges also contribute to deposition by cohesive attraction and are more relevant for the deposition in lower airways [80].

## 4. Techniques for Preparation of Dry Powders

It is challenging to prepare dry powder for inhalation, especially within the most desired particle size range of 1–5 µm [81,82]. Researchers have studied many techniques to achieve this ideal size range, including milling, freeze-drying, spray-drying, spray-freeze-drying, and supercritical fluid-drying. Recently, novel technologies such as particle replication in non-wetting templates (PRINT), inkjet-printing (IJP), thin-film freezing (TFF), and hot-melt extrusion (HME) have emerged as potential technologies for the preparation of improved dry powder for inhalation. Among all these mentioned techniques, milling and spray-drying are mostly used in pharmaceutical companies to prepare dry powder for inhalations (DPIs) [83,84,85,86,87,88]. Critical control parameters, advantages and disadvantages of all these techniques are compared in Table 1.

### 4.1. Milling

Milling is a traditional technique utilized for reducing particle size. This is a preferred method employed by pharmaceutical industries for optimization of pharmacological and physical properties of drugs, including solubility, stability, and bioavailability [89]. While jet-milling is comparatively cheaper and straightforward, there are several disadvantages to this technique. For example, particles produced by jet-milling are irregular in shape, have a rough surface, and have high levels of electrostatic charge on the surface [90,91]. In the process of jet-milling, coarse particles are subjected to high impacts with compressed air/gas, causing them to break into micro-sized particles, which are separated from larger particles by inertial impaction (Figure 3) [91]. This technique is suitable for thermolabile and meltable materials [92]. However, the particles generated by milling often exhibit poor flow properties compared to the parent coarse particles. It is thought that the high input energy required for milling creates a thermodynamically activated surface on the particles, which negatively affects their flow properties (Figure 3) [93]. Another problem with this milling technique is that particles display high levels of cohesive forces and are less effectively delivered from an inhaler device even though they have a smaller median size [94]. Despite the several drawbacks, the milling technique has contributed greatly to the development of dry powder for respiratory delivery. There are several DPI formulations reported prepared by using jet-milling techniques. Some formulations are listed below in Table 2.

### 4.2. Spray Drying

At present, the spray drying (SD) method is most commonly used for the preparation of dry powder for inhalation [52,95,96,97,98,99]. Operation of SD is relatively simple. In this method, liquid solution, suspension, or emulsions are sprayed like a mist through the nozzle into the drying chamber. The sprayed liquids encounter the hot air in the drying chamber and are dried into fine particles, further separated from the air in the cyclone, and ultimately collected in a collection chamber (Figure 4I) [100]. SD is a very convenient method for the preparation of dry powder for inhalation, as the particle size, size distribution, moisture content, and morphology of the particles can be controlled by optimization of several parameters such as solid content in the solution, solvent type, and instrumental conditions (i.e., solution feed rate, inlet temperature, gas supply and use of different types of nozzles). Particles obtained from SD methods are uniform in surface morphology, particle size, and size distribution with reasonable yield [101]. The solvent used in the formulation for spray drying plays an important role in the formation of particle size. Solvents with lower boiling points are easily evaporated and leave smaller-sized particles with increased yields [102]. Alobaidi et al. found that the use of two different solvents with low boiling points (acetone/methanol: 150/150) for spray dry of griseofulvin-PVP produced smaller particle size in comparison with the solvent combination of one polar and another non-polar (acetone/water: 185/85) [103]. Harjunen et al. found that lactose dried with 100% water and 100% ethanol was 100% amorphous and 100% crystalline, respectively [104]. Likewise, the feed rate has been observed to influence the physical parameters of dry powder. For example, slow feed rates result in smaller particle sizes with less moisture content and enhanced flow properties [105]. In contrast, a high feed rate usually results in particles with bigger size, higher moisture content, and poor dissolution rates [106]. The inlet temperature also has critical importance in the spray drying procedure by affecting the surface morphology, density, and water content of particles, as well as the overall product yield. Particles drying at low inlet temperatures have simultaneously higher water contents and poorer flow rates but have smoother surface morphologies. The effect of temperature has been well demonstrated in a study by Maas et al. [107]. In this study, the authors dried mannitol at three different temperatures (60 °C, 90 °C, and 120 °C) and compared the smoothness of the particles’ surface among the three temperature groups. Particles obtained at 60 °C and 90 °C were found to have smooth surfaces, whereas particles obtained at 120 °C were found to have rough surfaces. They also found that the particles obtained at higher temperatures were hollow in structure (Figure 4II). The effect of inlet temperature was also studied by Coppi et al., who prepared alginate microparticles loaded with lactate dehydrogenase and found that higher inlet temperatures decreased the water content in the dried powder while maintaining excellent storage stability [108]. Additionally, the effects of various instrumental parameters such as higher inlet temperatures on the product yield and stability during the drying process were also studied by Broadhead et al. [109].

Aperture size and operation mechanism of the spraying nozzle also play important roles in the determination of resulting particle size. Nozzle openings with bigger sizes spray large volumes of liquid, which increase the particle load in the drying chamber and cyclone into the collector chamber without completely drying. Nozzles are surrounded by a dry gas discharge vent, which helps evaporate the solvent from the sprayed liquid. Proper atomization of the nozzle openings and the gas supply are important during SD. The most used nozzle is the pneumatic nozzle, which is surrounded by a compressed gas vent. There are also other types of nozzles (Figure 3III) used in SD, such as pressure nozzles, rotary nozzles, ultrasonic nozzles, and four-fluid nozzle. Different types of nozzles require different sources of energy to operate. For example, pneumatic nozzles use compressed gases, rotary nozzles use centrifugal forces, and ultrasonic nozzles use ultrasonic energy to operate. The selection of nozzles is dictated by the desired size of powder production [18]. Recently, a more advanced four-fluid nozzle was used by Mizoe et al. to dry water-insoluble drugs Ethenzamide and Flurbiprofen. With the help of this nozzle, they passed two different drug solutions through two different nozzles and carrier gas through the other two different nozzles [110]. Some dry powders for inhalation prepared by the spray drying technique are listed in Table 2.

### 4.3. Spray-Freeze-Drying

Like conventional spray drying, the spray-freeze-drying (SFD) technique is also used for the preparation of dry powders for inhalation [97]. Although this method is relatively complicated and costly compared to conventional spray drying, thermolabile drugs are often prepared by this technique. Powder for inhalation is obtained through SFD in two steps. In the first step, nano- or micro-particle suspension is sprayed on the surface or in the bed of liquid nitrogen under a controlled supply of compressed gas (Figure 5I,II). The second step involves freeze-drying [91,111] to get dry powder. The principle of SFD is simple: after the liquid spray is sprayed, the droplets come into contact with liquid nitrogen (−195.79 °C) and rapidly (in milliseconds) solidify due to the high heat-transfer rate [112]. To prevent the formation of agglomerates, the liquid nitrogen is kept on mild stirring. After completion of the spraying procedure, the resulting mass is vacuumed to remove the water content and nitrogen vapor [113,114]. The desired particle size can be obtained by optimization of the spray feed rate and the solid content ratio in the feed liquid [115]. SFD is a combination of SD and freeze-drying (FD) processes. To produce different particle sizes, different atomizers can be used to spray the liquid and to dry frozen particles. The sublimation process is utilized similarly as in FD [116]. SFD results in particles with large sizes (>6 µm) and with porous morphologies that are more suitable for aerosolization in comparison to those obtained from SD, which are usually smaller in size (<3 µm). Due to larger particle sizes and higher porosity, SFD particles have enough space to aerosolize [117,118]. A very good example of the spray freeze-drying process was reported by Kondo et al., who prepared spherical and porous tolbutamide-loaded hydroxy propyl methyl cellulose (HPMC) particles using SFD [23].

Although the particles obtained from SFD have uniform particle size and size distribution with low bulk densities, which are both important parameters for pulmonary application, this method is not well developed and therefore is not suitable for large-scale production [31,97,119]. As this method does not use heat for drying, it is much more commonly used in the production of heat-sensitive products such as proteins, monoclonal antibodies, vaccines, enzymes, and plasmids [112,117,120,121,122]. Some researchers have reported that many proteins undergo denaturation during spraying through a tiny nozzle hole [115,123,124]. However, applications of stabilizers, including mannitol [125], lactose, trehalose [126], zinc [127], and surfactants like polysorbate-80 [128,129] have been shown to protect proteins from denaturation by reducing the friction during spraying and forming a protective layer on the droplets [127,130]. To protect the original state of the drug during SFD, a novel large porous particle (LPP) technique was discussed by Ogienko et al. [131]. In this study, this group used glycine as a carrier to form LPP to deliver slabutamol and budesonide as model inhalable drugs that have poor water solubility. Some dry powders for inhalation prepared by SFD technique are listed in Table 2.

### 4.4. Supercritical Fluid Drying

Supercritical fluid (SF) drying is a new technology for the preparation of micro-sized particles for pulmonary delivery [97]. Supercritical condition is a thermodynamic condition for any chemical substance, in which, when temperature (T) and pressure (P) exceed their critical values, *Tc* and *Pc* respectively, the substance remains neither in liquid nor in gas state and behaves as both a liquid and a gas (Figure 6I) [132]. The density of SF is like that of liquids with high compressibility and intermediate viscosity and diffusivity. These properties of SF are the main driving properties that help to precipitate solid mass from liquid solution in the medium. The high compressibility property can help in mass transfer and can be controlled by varying the temperature and pressure. Similarly, the viscosity and diffusivity properties facilitate the solvation capacity of the SF [26]. Carbon dioxide (CO_2_) is most widely used in pharmaceutical preparation as an SF as its advantages include (1) low critical temperature (31.1 °C) and moderate pressure (73.8 bar), (2) non-toxicity and non-reactivity, and (3) low production costs [91,133,134]. However, the use of CO_2_ as SF contains drawbacks such as limited solvation power for some compounds [26]. However, its solvation power can be enhanced by the addition of some organic solvents like ethanol or acetone [135]. This SF has been used for particle size reduction of chemicals in many sectors like cosmetics, paint, and pharmaceuticals. This technique is further classified into different categories based on the SF used, for example, rapid expansion of SF (RESF), gas antisolvent (GAS), aerosol solvent extraction system (ASES), and solution-enhanced dispersion of solids (SEDS). Among these techniques, the RESF is mostly used to prepare organic solvent-free fine dry particles [136]. In this technique, the SF acts as a continuous phase. The solid materials to be micronized are first solubilized in SF and then have pressure applied to them to expand the solution. The rapid expansion in the SF can be depressurized by passing through a heated nozzle which causes rapid nucleation of the substrate, resulting in very fine micronized particles [137]. In the RESF-CO_2_ technique, CO_2_ is passed into the reactor tank, where the temperature and pressure of the CO_2_ are maintained above their critical point and sample solution is sprayed into the SF-CO_2_. The feed rate pressure of the CO_2_ is maintained in the reactor tank throughout the process by a high-pressure pump, and the temperature in the reactor tank is maintained by the circulation of water in the outside jacket of the tank (Figure 6II). Upon expansion under pressure and high temperature, the sprayed sample in the SF-CO_2_ precipitate can be separated by venting out SF-CO_2_. Particles obtained by this technique have uniform size and size distribution with amorphous morphology and enhanced dissolution rates.

Many pharmaceutical products have been prepared for pulmonary delivery by using SF-CO_2_. For the treatment of asthmatic conditions, Rehmanet prepared Terbutaline sulfate with improved fine particle fraction and surface morphology by using SF-CO_2_ [138]. Similarly, a metered-dose inhaler of fluticasone-17-propionte was prepared with SF-CO_2_ by Steckel [139]. Likewise, β_2_-adrenergic bronchodilator salmeterol xinafoate was prepared by this method with improved aerosol properties for the treatment of bronco-congestion [140]. Furthermore, dry powder of ibuprofen-loaded chitosan microparticles for inhalation with an aerodynamic diameter of 1.21 µm was prepared using SF-CO_2_ by Cabral et al. [141]. Recently, SF has become popular in the biopharmaceutical sector for the preparation of protein and nucleic acid dry powders for inhalation. Many proteins are hydrophilic and are not freely soluble in SF, which serves as an anti-solvent. Proteins to be precipitated are first dissolved in a suitable solvent, which is miscible with SF, after which the SF is passed into the protein solution to reduce the protein solubility and precipitate it into fine microparticles [2]. Thiering et al. used organic and aqueous solvents to dissolve lysozyme, insulin, and albumin, where CO_2_ and ammonia were used as anti-solvents to precipitate these proteins in order to obtain dry powders for inhalation with aerodynamic diameters between 0.05–2 µm [142]. This SF-CO_2_ was used as an anti-solvent by Douglas et al. to produce recombinant human immunoglobulin dry powder for inhalation [143]. Similarly, recombinant human growth hormone was precipitated from aqueous solution by SF-CO_2_ in the presence of isopropanol as co-solvent, and sucrose was used to prepare dry powder with aerodynamic diameter 1–6 µm for pulmonary administration [144]. Nucleic acids have been successfully prepared as dry powders by using enhanced dispersion with the SF method. As nucleic acids are not stable in liquid form and are prone to degradation during administration, a stable dosage form is required. A comprehensive study was performed by Tservistas et al. [145] regarding the availability and stability of radio-labeled plasmid DNA expressing chloramphenicol acetyl transferase in the lungs of mice. They reported that optimal concentration of plasmid DNA was not achieved in the lungs of mice even after 2 h post-intravenous injection, and the majority of plasmid DNA was found degraded in blood circulation. In an attempt to improve the bioavailability of plasmid DNA, the authors prepared plasmid DNA-loaded mannitol dry powder for pulmonary delivery. They used the SF-CO_2_ technique to prepare the dry powder in presence of isopropanol as a co-solvent. Likewise, Okamato et al. prepared a chitosan-plasmid DNA complex as dry powder with the application of SF-CO_2_ and ethanol as co-solvent [146]. Furthermore, the SF-CO_2_ technique has also been used for the preparation of live-attenuated measles vaccines [147,148] and dry short interfering RNAs [149] for pulmonary administration. Some of the pharmaceutical formulations for inhalation prepared by this technique are listed in Table 2.

### 4.5. New Emerging Technologies

Recently, novel technologies such as particle replication in non-wetting templates (PRINT), inkjet-printing (IJP), thin-film freezing (TFF), and hot-melt extrusion (HME) are emerging as potential technologies for the preparation of improved dry powder for inhalation. PRINT is soft lithography techniques that use perfluoropolyether elastomers as a molding template on a silicone master plate to create different shaped micro to nano size particulate matters. Recently, some research groups have used this technology to improve the size and flow property of dry powder for inhalation. Garcia et al. have used this PRINT technology to prepare zanamivir-loaded microparticles, which showed 3.19-fold improved flow properties in comparison to conventional DPI technologies [150]. Likewise, IJP is another novel technology that can precisely control the morphology of particles with a digital imaging system. In this technology, the processing liquid materials can be impelled dropwise on suitable substrates with defined particle size and morphology [151]. Lopez-Iglesias et al. have prepared IJP-based salbutamol sulfate-loaded alginate aerogel microspheres and demonstrated that these particles were highly porous in the range of 2.4 µm with an improved FPF (49.7%) compared to the powder prepared by conventional technology [152]. This group has mentioned that this IJP technology can be further utilized to design personalized aerosols with improved FPF and MMAD of powder. TFF is a freezing technology, where the freezing of liquid is controlled under the influence of a fluid dynamic system. In this process, liquid formulations are rapidly spread in the form of thin film on a cryogenically cooled surface, where the transfer of heat from the spread liquid droplets takes place within a fraction of second to convert it into solid mass, which is further lyophilized to get dry powder. Various research groups have mentioned that dry powders prepared by this TFF technology have low bulk density, smaller size, and a good respirable property [153,154]. Sahakijpijam et al. have prepared TFF-based tacrolimus DPIs and demonstrated lower MMAD and higher delivered dose [154]. HME is used in the pharmaceutical industry to enhance the solubility of low soluble drugs, mask the unpleasant taste of drugs, and formulate prolonged drug delivery formulations. In this technology, a solid mixture of drug/polymers is heated together into a mold beyond its glass transition temperature (Tg) to melt into a viscous mass, which is further collected as a slug to micronize as a fine powder. Lin et al. prepared HME-based itraconazole inhalable powder. This group initially jet-milled itraconazole with mannitol (20:80) and then extruded it through twin-screw extruder to generate slugs which were further passed through the jet-mill to obtain inhalable-size powder (2.19 µm) [155].

## 5. Factors Influencing the Clinical Efficacy and Marketed Formulations of DPIs

The clinical efficacy of DPIs depends on three major factors, i.e., types of formulations, the device used to deliver the formulation, and the patient who uses the device. As discussed above, physico-chemical properties of the drug substance such as solubility, particle size, morphology, and preparation method are important factors to be considered while formulating dry powder for inhalation. Particles of size ≤0.5 µm may be exhaled out or quickly absorbed into the systemic circulation following alveoli deposition, and particles >5 µm can be easily deposited in the oropharynx and may never reach the lung. The critical factors affecting lung deposition of the delivered powder are FPD, FPF, and MMAD. Aerosolized powders with high FPF, FPD, or low MMAD are more likely deposited in the deep lung.

There are various types of DPI devices developed by different pharmaceutical companies to deliver DPI formulations effectively. All these devices are different in operation, but the working mechanism is the same, i.e., passive. This means it depends on the patient’s breath to activate the drug delivery; it is therefore important to know about the patients’ conditions before prescribing the formulations. These DPI devices have different degrees of internal resistance airflow, which can be classified by the inhalation flow required to produce a 4 kPa pressure drop [216]. Janson et al. have reported an in vitro comparative study of FPD delivered from three different DPI devices, i.e., Turbuhaler^®^, Spiromax^®,^ and Easyhaler^®^ with the same formulation (budesonide/formoterol) [217]. The FPD ratios of low vs. medium flow and high vs. medium flow were similar for all devices and strengths and for both components. The FPD for the budesonide component was consistent from all devices but for formoterol was consistently higher with Turbuhaler^®^ compared with the other devices. The authors have concluded that the devices tested were equally flow-dependent with regards to the FPD of budesonide and formoterol. However, as the dependency of FPD on the inhalation flow rate is a critical parameter of DPIs, it was noted that the magnitude of decrease in FPD for some of the devices tested may have clinical implications in patients with low inhalation capacity. There are some other DPI devices reported, such as HandiHaler^®^, Breezhaler^®,^ and Diskus^®^; delivery of dry powder from these devices depends on the inhalation volume capacity of the patients [218,219]. Assessments with the Diskus^®^ device showed higher inspiratory volume capacity when used in healthy volunteers than in diseased patients with asthma, COPD, or neuromuscular disease [220].

The clinical efficacy of DPIs also depends on the patient’s health condition, age, and sex [221,222,223]. In an observational study, Melani et al. showed that incorrect use of DPIs is widely distributed. In older (>60 years) patients, incorrect handling of DPIs is more frequent than in younger patients [223]. In another observational study, it was found that asthma patients have difficulty in the proper handling of DPI devices [224]. Therefore, the prescriber must assess the patient’s conditions, peak inspiratory flow, and inhalable volume before prescribing DPIs. For patients with physical or cognitive impairment, DPI with fewer operation steps should be chosen. Once-daily dosing may be better for these patients [221].

From the above factors, the prescriber physician must be familiar with the DPI formulations and various devices that directly affect the delivery of dry powder and influence the clinical outcome. For patients with poor lung function, certain types of DPIs may not be the most suitable choice, due to their varying internal resistance to airflow.

As mentioned above, the pulmonary route of drug administration has shown various advantages over oral and parenteral routes of drug administration. Recently, various research studies are going on to develop effective pulmonary formulations (DPIs, pMDI, and nebulizer) to target local pulmonary as well as systemic diseases. It has been reported that about 75% of drugs are under research and development, and about 40% of pulmonary formulations are in market for the treatment of various pulmonary as well as systemic diseases [225]. Marketed DPIs formulations for clinical use are listed in Table 3.

## 6. Challenges and Future Perspectives

Despite improvements in the formulation of dry powder and preparation techniques, delivery of therapeutics through the respiratory route is still far from perfect. Several challenges need to be addressed. The delivery of dry powder in the form of aerosol depends mostly on three factors: architecture of the respiratory system, physical properties of dry powder, and devices used for inhalation of such aerosols. Among these three factors, the architecture of the respiratory system and physical properties of dry powder play a major role in aerosolization and deep lung deposition. It is thought that devices have minimal influence on the bioavailability of the inhaled drug. However, the proper handling of such devices determines the bioavailability [226]. Discussion on the role of devices in aerosolization is beyond the scope of this review article; however, readers are encouraged to see an excellent review on this topic by Zhou et al. [227]. Inhaled particles pass through a long tracheal path, where they encounter many intrinsic and extrinsic factors, such as a moist environment, the presence of mucus inside the tracheal path, gravitational force, active metabolic barriers, and macrophages. Approximately 50–60% of delivered particulate matters are deposited into the pharyngeal cavity [33] and may cause local or systemic adverse effects. Nearly 100% humidity in the tracheal trajectory [228] can rapidly hydrate the inhaled powder and increase the bulk density, thereby leading to sedimentation in the tracheal path. Meanwhile, anatomical structures, such as the presence of mucus, gradual decrease in diameter, and irregular branching of trachea to bronchi, reduce the movement of respired matters. The windpipe in the respiratory tract from the main conducting zone to alveoli is lined with respiratory epithelium, which contains cilia surrounded with mucus. This mucociliary mechanism helps to maintain epithelial moisture and traps inhaled particulate materials to clear from airways via coughing [229]. Furthermore, the presence of active metabolic enzymes and alveolar macrophages at every region of the lungs poses challenges in the optimal delivery of therapeutics. The metabolic enzymes secreted by alveolar macrophages, lymphocytes, neutrophils, and mast cells are responsible for the degradation of inhaled particulate matters. The alveolar macrophages, which provide a first line of defense in the lungs are of major concern due to their high density [37]. These macrophages are 15–50 µm in diameter and reside in contact with the surfactant lining of the alveoli. In the lungs, foreign particulate matters encounter alveolar macrophages simply by electrostatic force or by receptor-mediated interaction. Once the particle comes into contact, they are engulfed by alveolar macrophages and migrate to the ciliated epithelium for clearance. These mechanisms of clearance reduce the bioavailability of inhaled drugs. Multiple administrations of a drug, up to 3–4 times per day, may saturate the macrophages and hence result in effective absorption of the drug from the lungs. However, the multiple-administration approach is not only less desirable from the perspective of patients’ compliance but also increases the risk of off-target toxicities. Our knowledge about particle deposition and the complicated respiratory physiology of lungs is still sub-optimal, and continued research work in this field will enhance the overall safety and efficacy of inhaled therapeutics.

As discussed above, physical properties of particles such as size, shape, density, surface charge, and moisture content directly influence the aerosolization of dry powder. Therefore, many efforts have been made to improve the physical properties of dry powder in order to enhance their aerosolization properties. It has been well documented that particles with diameters of 1–5 µm and bulk density <1 g/cm^3^ are optimal for deep lung deposition. However, it has also been reported that particles with a size of 1–5 µm are more susceptible to phagocytosis by alveolar macrophages. Many modifications in formulations have been studied in order to preclude recognition and clearance of particles by alveolar macrophages. Such modifications, as discussed in earlier examples, include increases in size, changes in particle’s surface morphology, and pegylation. The safety of these modified particles is a concern, especially when an extra molecule such as polyethylene glycol is added during the modification process.

As mentioned above, the most frequently used techniques in pharmaceutical industries to prepare dry powder for inhalation are milling and spray drying. Besides these, there are several techniques such as freeze-drying, spray-freeze drying, supercritical fluid technique, PRINT, IJP, TFF, and HME are used for the preparation of dry powder for experimental purposes. These techniques have their own advantages and disadvantages as discussed above. For example, milling is a common and cheap technique to produce fine particles within a respirable range. However, the resulting particles often have altered physicochemical properties due to the high input impaction forces from applied air during micronization. These milled particles are partially amorphous, with high levels of electrostatic surface charges. Due to the strong presence of surface charge, large energy inputs are needed to disperse the powder during inhalation. The spray-drying technique utilizes high temperatures to dry the spread liquid in the drying chamber and convert it to dry powder. Thermo-degradable drugs are not suitable for this technique. Spray-freeze-drying and supercritical fluid techniques are suitable for thermo-degradable drugs, and the obtained dry powders have suitable physicochemical properties, but these processes are expensive in comparison to milling and spray-drying. Despite advancements in engineering techniques to produce dry powder within respirable range, adequate stability, flowability and dispersibility are still challenging to address.

Although formulation-based knowledge for the preparation of dry powder for inhalation has been markedly increased, there are still many misconceptions. These include the beliefs that high-resistance devices are unable to deliver enough powder and are not suitable for all patients, that the extra-fine powder (<1 µm) particles can improve peripheral lung deposition, and that inhalers with flow-rate-independent fine particle fractions produce a more consistent delivery to the lungs. Another misunderstanding is about the rapid clearance of the inhaled particles from the lungs.

Taken together, there are several issues that need to be addressed in order to achieve optimal drug delivery with minimal adverse effects via the respiratory route. Although there have been advancements in knowledge about the respiratory system and pulmonary drug formulations, more research is warranted to bring the respiratory route of drug delivery to the forefront.

## 7. Conclusions

The pulmonary route has been used for drug delivery since ancient times due to its many advantages over other routes of drug administration. It is preferred for drug administration mainly due to the large absorptive surface area of the lungs and the bypass of the hepatic portal system. Previously, the pulmonary route was mostly utilized for the treatment of local respiratory diseases such as asthma, chronic obstructive pulmonary disease, and cystic fibrosis. With advancements in drug formulation techniques, this route is gaining popularity for the treatment of systemic diseases such as diabetes. Recently, medications for Parkinson’s disease, Alzheimer’s, and lung cancer have also been administered through the pulmonary route. Despite various advantages of this route of drug administration, it is still technically challenging to develop successful formulations. Furthermore, the complex physicochemical properties of dry powders as well as the complex geometry and architecture of lungs also pose difficulties in the successful formulations of drugs to be used via inhalation, as discussed in this review. The development of particles with optimized physicochemical properties would ensure the successful aerosolization and deposition of dry powder in the deep lungs. The selection of suitable techniques for the preparation of particles is also important, as mentioned in this review. Despite technical challenges, this field has recently gained momentum, and further improvements are expected with a better understanding of the physicochemical properties of particles and biology of the respiratory system.

## Figures and Tables

**Figure 1 pharmaceutics-13-00031-f001:**
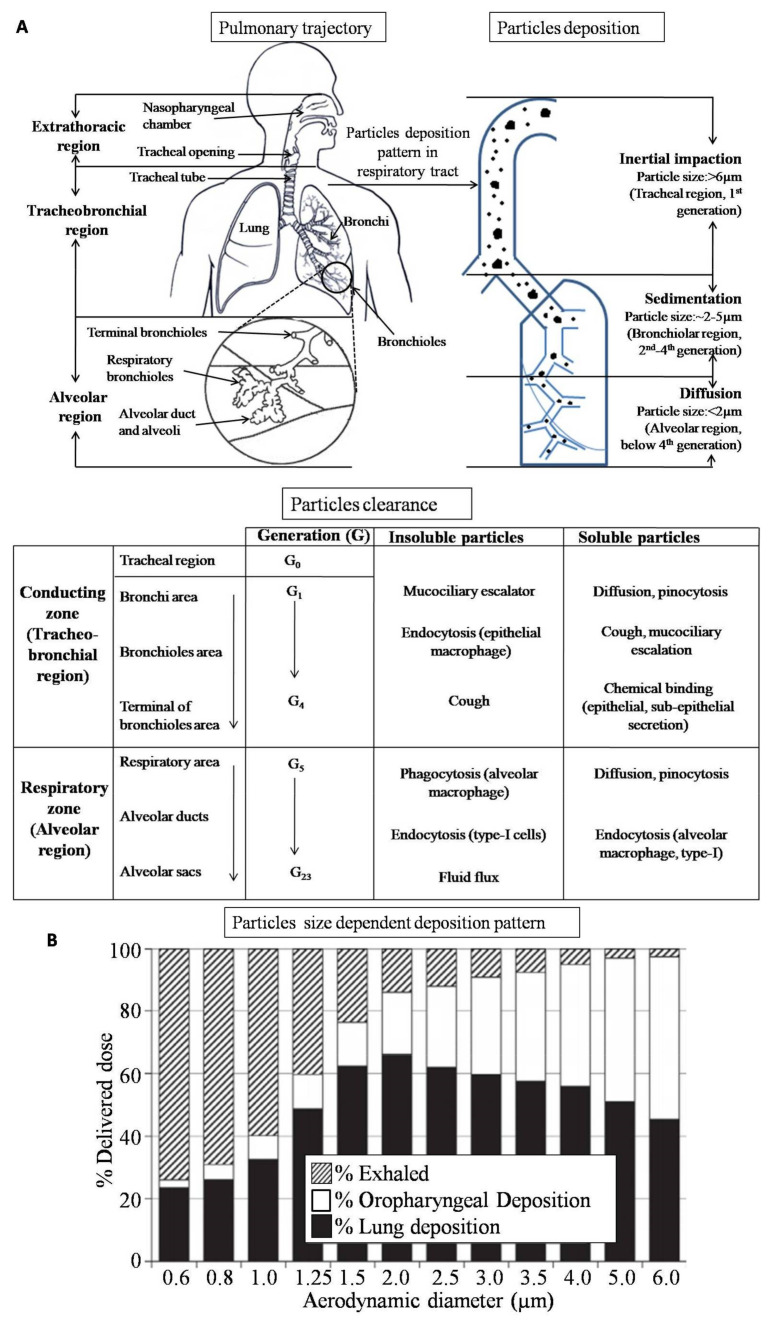
Airways system and particle deposition pattern. (**A**) Pulmonary tract, particle deposition pattern, and clearance mechanism. (**B**) Size-dependent deposition of particles in the respiratory tract Reproduced with permission from [44], Elsevier, 2015.

**Figure 2 pharmaceutics-13-00031-f002:**
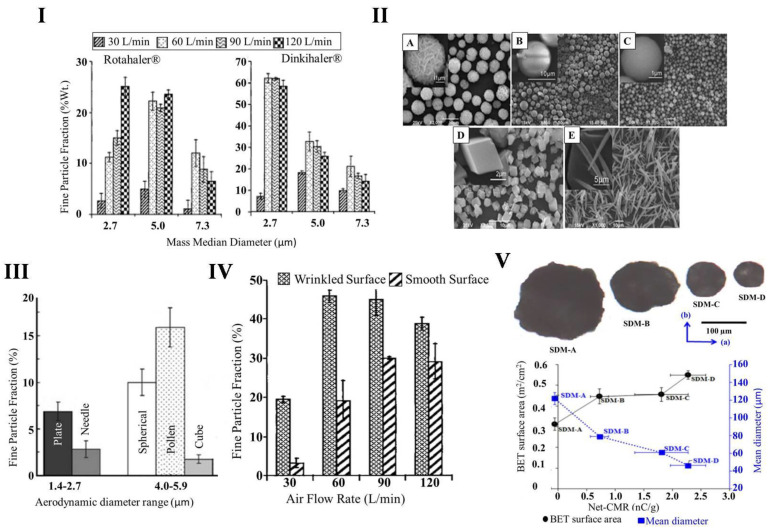
Physicochemical properties of dry powder for inhalation; (**I**) Effect of particle sizes on fine particle fraction (FPF) aerosolized at different air volume. Reproduced with permission from [52], Elsevier, 2000; (**II**) SEM images of differently shaped particles prepared for aerosolization: (**A**) pollen-shaped, (**B**) spherical shaped, (**C**) plate-shaped, (**D**) cube-shaped, and (**E**) needle-shaped particles; Reproduced with permission from [53], Springer, 2009; (**III**) the effects of shape on aerosolization. Reproduced with permission from [53], Springer, 2009; (**IV**) comparison of dispersibility of wrinkled and smooth-surfaced particles at different airflows. Reproduced with permission from [14], Mary Ann Liebert, 2002; (**V**) net surface charge (nC/g = nanocoulomb/gram,) in relation to volume mean diameter of spray-dried mannitol (SDM-A_90 µm_, B_63 µm_, C_45 µm_, and D_20 µm_) Reproduced with permission from [70], Springer, 2013.

**Figure 3 pharmaceutics-13-00031-f003:**
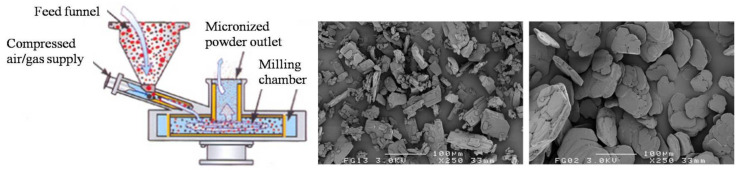
Preparation of dry powder by jet milling. Jet mill and its component (**left**); scanning electron microscope image of jet-milled dry powder (**right**).

**Figure 4 pharmaceutics-13-00031-f004:**
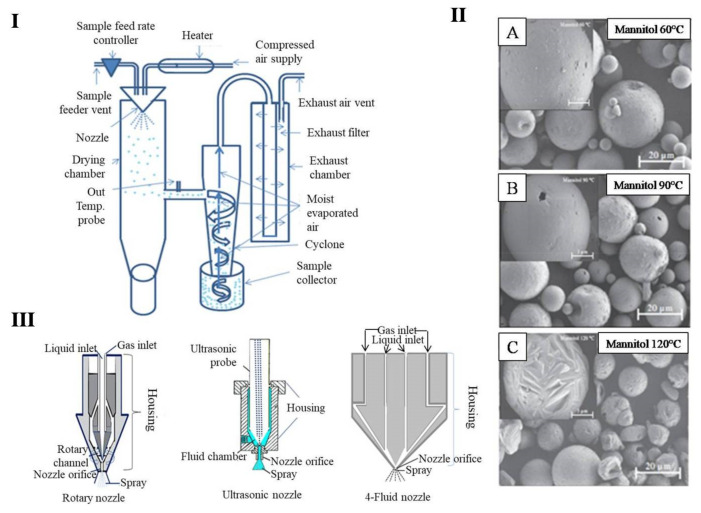
Spray drying technique. (**I**) Schematic diagram of spray dryer and its component; (**II**) scanning electron microscope image of spray-dried mannitol, (**A**) at 60 °C, (**B**) at 90 °C, and (**C**) at 120 °C of outlet temperature. Reproduced with permission from [107], Elsevier, 2011; (**III**) schematic diagram of different nozzles.

**Figure 5 pharmaceutics-13-00031-f005:**
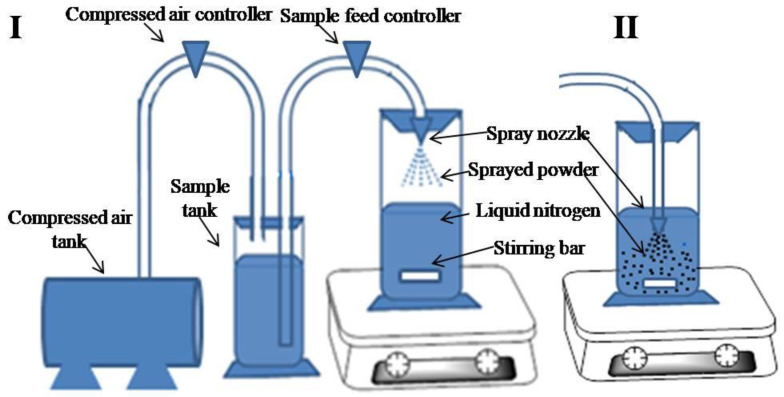
Schematic diagram of spray freeze drying process; (**I**) spray liquid in the liquid N_2_ vapor, (**II**) spray of liquid in the bed of liquid N_2_.

**Figure 6 pharmaceutics-13-00031-f006:**
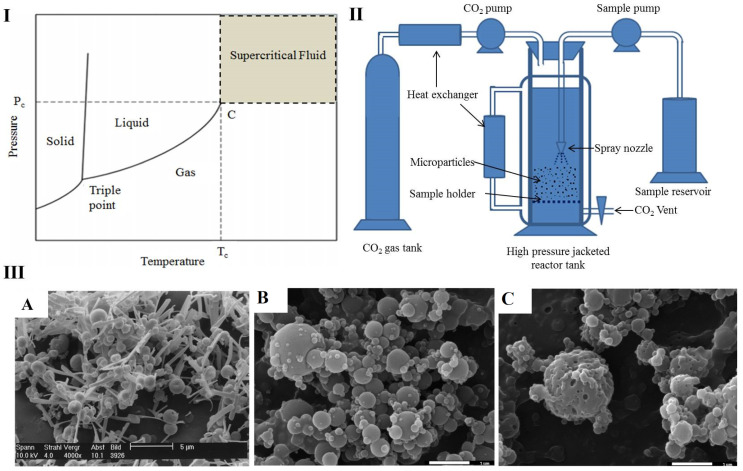
Preparation of dry powder for inhalation by supercritical fluid drying. (**I**) Chemical substance phase diagram (C: critical point) Reproduced with permission from [132], Elsevier, 2015; (**II**) schematic diagram of rapid expansion of supercritical fluid (RESF)-CO_2_ technique; (**III**) SEM images of some supercritical fluid (SF)-dried products (**A**) fluticasone with 5% lecithin Reproduced with permission from [139], Elsevier, 1998, (**B**,**C**) CHT and CHT-IBP microparticles Reproduced with permission from [141], Elsevier, 2016.

**Table 1 pharmaceutics-13-00031-t001:** Control parameters, advantages, and disadvantages of various techniques.

Techniques	Control Parameters	Advantages	Disadvantages
**Milling**	Force of inlet air/gas Moisture content Milling environment Feeding materials physical and chemical properties.	Simple, easy to handle, and inexpensive technique Well-established and validated method Suitable to reduce particle size within an inhalable range	Not suitable for fragile materials because high force is required to mill the materials, which can possibly contribute to chemical and physical instability. Micronized powder has poor flow property due to the generation of high surface electrostatic charge. Micronized powder has irregular size and shape Mostly micronized powders are amorphous
**Spray-drying**	Viscosity of feeding liquid Solute content in feeding liquid Inlet and outlet temperature Feed flow rate Atomization airflow Spraying nozzle size	Easy to scale up. Single-step process, reproducible, and economical. Robust technique; always produce uniform powder at selected parameters. Obtained dry powders are within a suitable inhalable range. Suitable for both organic and aqueous solvent-based drugs. Mostly drugs converted into dry powder through this technique, maintain their physical as well as chemical properties.	Not suitable for heat-sensitive formulations Low yield
**Spray-freeze-drying**	Solute content in feeding liquid Feeding flow rate Atomization airflow Spraying nozzle size	Suitable for heat-sensitive formulations, which are not suitable for spray drying. Mainly suitable for biological formulations. Obtained dry powders have low density and are often porous. Always final products obtained with high yield.	Complex and time-consuming technique. Comparatively costly technique. Not suitable for the formulations that do not withstand cryogenic stress and share stress during atomization.
**Supercritical fluid drying**	Solute content in liquid Feed flow rate Type and flow rate of co-solvent Tank pressure Atomization airflow and nozzle size	Suitable for biological formulations. Rapid process. Obtained powders are spherical and uniform with a smooth surface.	Highly expensive and special set-up required. Exposure to organic solvent.

**Table 2 pharmaceutics-13-00031-t002:** Some of the dry powders for inhalation prepared by various techniques.

Drug/Payload	Additives	Median Size	Ref.
**A. Milling**			
1. Recombinant secretory leukocyte protease inhibitor	1,2-Dioleoyl-sn-glycero 3[phosphor-Lserine], Cholesterol	2.44 µm	[156]
2. β-Glucuronidase	Dimyristoylphosphatyl-Choline, Cholesterol	6.4 µm	[157]
3. Beclomethasone dipropionate	Not mentioned	~5 µm	[94]
4. Levodopa	L-Leucine	<5 µm	[158]
5. Fluticasone-17-propionate	HPMC	~2 µm	[94]
6. Fusafungine	Lactose	~5 µm	[159]
7. Diclofenac	Not mentioned	2.36 µm	[160]
8. Simvastatin	Not mentioned	2.2 µm	[161]
9. Itraconazole	Mannitol and Sodium taurocholate	5.91 µm	[162]
10. Indomethacin	Mannitol and L-leucine	0.96 µm	[163]
11. Glucagon	Pharmatose, Erythritol	~2.5 µm	[164]
12. Glucagon	Citric acid, Lactose	4.7–52.1 µm	[165]
13. Ciprofloxacin HCl and Colistin sulfate	Not mentioned	<5.4 µm	[166]
14. Salbutamol sulphate	Not mentioned	~10 µm	[167]
**B. Spray drying**			
1. N-acetylcysteine	Soya phosphatidylcholine, Cholesterol, Polysorbate 80	2.72 µm	[168]
2. Dapsone	Dipalmitoylphosphatidylcholine, Cholesterol, Polysorbate 80	2.2 µm	[169]
3. Rifampicin	Soya phosphatidylcholine, Cholesterol, Hydrogenated soybean phosphatidylcholine	~2 µm	[170]
4. Rifapentine	Not mentioned	1.92 µm	[171]
5. Isoniazide	L-α-soybean phosphatidylcholine, Cholesterol, Mannitol	4.92 µm	[172]
6. Ciprofloxacine	Hydrogenated soybean phosphatidylcholine, Cholestrol, Sucrose	~1 µm	[173]
7. Tacrolimus	Hydrogenated soybean phosphatidylcholine, Cholesterol, Trehalose	2.2 µm	[174]
8. Docetaxel	Phosphatidylcholine, Cholesterol, Mannitol, Leucine	3.1 µm	[175]
9. Amiloride HCl	Hydrogenated soy phosphatidycholine, Cholesterol, Mannitol	2.3 µm	[176]
10. Moxifloxacin	Phosphatidylcholine, Cholesterol, Dextran	<5 µm	[177]
11. Oseltamivir phosphate	Ovelecithin, Cholesterol, Leucine	~3.5 µm	[178]
12. Salmon calcitonin	Sodium tripolyphosphate, Chitosan, Mannitol	4.7 µm	[179]
13. Azethromycin	Not mentioned	1.6 µm	[180]
14. Paclitaxel	Dipalmitoylphosphatidylcholine, dipalmitoylphosphatidylglycerol	2.3 µm	[181]
15. Tobramycin	Poly(lactic-co-glycolic acid), Poly(vinyl alcohol)	3.3 µm	[182]
16. Tobramycin (PulmoSphere™)	Distearoylphosphatidlcholin, perflurooctyl bromide	~5 µm	[183]
17. Zanamivir (Relenaza^®^)	Mannitol, L-leucine, Poloxamer 188	2.3 µm	[184]
**C. Spray-freeze drying**			
1. Insulin	Soya lecithin, Cholesterol, Cholate, Mannitol	3.9 µm	[185]
2. Theophylline anhydrate and oxalic acid	Not mentioned	3.0 µm	[186]
3. Ciprofloxacin	Dimyristoylphosphatidylglycerol, lactose	2.8 µm	[187]
4. Levofloxacin	Polycaprolactone, L-leucine, Mannitol	~4–5 µm	[188]
5. Levofloxacin	Soybean lecithin, D-mannitol, L-leucine	5.6 µm	[189]
6. Small interfering RNA	Mannitol	10–14.9 µm	[190]
7. Voriconazole	Mannitol	3.8 µm	[191]
8. Octreotide acetate	Mannitol, ammonium carbonate	2.6 µm	[192]
9. Human IgG	Hydroxypropyl β-cyclodextrin, trehalose	~5.32 µm	[193]
10. Humanized anti-IgE monoclonal antibody	Carbohydrate excipients	~3 µm	[117]
11. PlasmidDNA-Luc	Β-benzyl-L-aspartate N-carboxy-anhydride	7.6 µm	[194]
12. Viral protein (hemagglutinin)	Dextran, Mannitol, Poloxamer 188, Polysorbate 20, Trehalose	30–60 µm	[195]
13. Monovalent influenza subunit hemagglutinin	Inulin	11.05 µm	[196]
14. Δ9-Tetrahydro-cannabinol	Inulin	84.1 µm	[197]
**D. Supercritical fluid drying**			
1. Terbutaline sulphate	α-Lactose monohydrate	2.85–3.43 µm	[138]
2. Ipratropium bromide	Bovine serum albumin	1–5 µm	[198]
3. Fluticasone-17-propionte	Poloxamer 188	~1.69 µm	[199]
4. Salmeterol xinafoate	Not mentioned	Not mentioned	[140]
5. Salbutamol sulphate	N-methyl 2-pyrrolidone	1–3 µm	[200]
6. Albuterol sulfate	α-lactose monohydrate	2.4 µm	[201]
7. Beclomethasone-17,21-dipropionate	Not mentioned	7.9 µm	[202]
8. Miconazole	Phosphatidylcholine, Cholesterol, Poloxamer 407	3.6–9.4 µm	[203]
9. Salmon calcitonin	Inulin, Trehalose, Chitosan, Sodium taurocholate, β-cyclodextrin	2.2–2.9 µm	[204]
10. Rifampicin	Poly(L-lactide)	<5 µm	[205]
11. Amoxicillin trihydrate	Not mentioned	Not mentioned	[206]
12. Piroxicam	β-Cyclodextrin	Not mentioned	[207]
13. Ibuprofen	Chitosan	2.1–2.7 µm	[141]
14. Insulin	Not mentioned	2–3 µm	[208]
15. Plasmid pSVβ	Mannitol	Not mentioned	[145]
16. Plasmid pCMV-Luc	Chitosan, trehalose	<10 µm	[209]
17. Plasmid DNA	Poly(D,L-lactic-co-glycolic) acid	Not mentioned	[210]
18. siRNA	Chitosan	<10 µm	[211]
19. 5-fluorouracil	α-lactose monohydrate	Not mentioned	[212]
20. Curcumin	Hydroxypropyl-β-cyclodextrin	~5.8 µm	[213]
21. Nalmefene hydrochloride	Not mentioned	0.5–2 µm	[214]
22. Cyclosporine A	Not mentioned	<2.5 µm	[215]

**Table 3 pharmaceutics-13-00031-t003:** DPI formulations available in the market of the USA and other countries.

Drug	Additives	Product	Manufacturer	Indications
Albuterol sulfate	Lactose monohydrate	ProAir Respiclick	Teva	Asthma and COPD
Salbutamol sulfate	Lactose monohydrate	Pulvinal Salbutamol	Chiesi	Asthma and COPD
Salbutamol sulfate	Lactose monohydrate	Easyhaler Salbutamol Sulfate	Orion	Asthma and COPD
Terbutaline sulfate	N/A	BricanylTurbohaler	AstraZeneca	Asthma and COPD
Salmeterol xinafoate	Lactose monohydrate	Serevent Diskus	GlaxoSmithKline	Asthma and COPD
Formoterol fumarate	Lactose monohydrate	ForadilAerolizer	Novartis	Asthma and COPD
Formoterol fumarate	Lactose monohydrate, Magnesium stearate	ForadilCertihaler	Novartis	Asthma and COPD
Formoterol fumarate	Lactose monohydrate	OxisTurbohaler	AstraZeneca	Asthma and COPD
Formoterol fumarate	Lactose monohydrate	Easyhaler Formoterol	Orion	Asthma and COPD
Indacaterol maleate	Lactose monohydrate	ArcaptaNeohaler	Novartis	Asthma and COPD
Tritropium bromide	Lactose monohydrate	Spiriva Handihaler	Boehringer Ingelheim	Asthma and COPD
Aclidinium bromide	Lactose monohydrate	TudorzaPressair	Forest	Asthma and COPD
Glycopyrronium bromide	Lactose monohydrate, Magnesium stearate	SeebriBreezhaler	Novartis	Asthma and COPD
Umeclidinium	Lactose monohydrate, Magnesium stearate	Incruse Ellipta	GlaxoSmithKline	Asthma and COPD
Budesonide	Lactose monohydrate	Easyhaler Budesonide	Orion	Asthma and COPD
Budesonide	Lactose monohydrate	Pulmicort Flexhaler	AstraZeneca	Asthma and COPD
Mometasone furoate	Lactose anhydrate	Asmanex Twisthaler	Merck	Asthma and COPD
Beclomethasone dipropionate	Lactose monohydrate, Magnesium stearate	PulvinalBeclometasone dipropionate	Chiesi	Asthma and COPD
Beclomethasone dipropionate	Lactose monohydrate	EasyhalerBeclometasone	Orion	Asthma and COPD
Fluticasone propionate	Lactose monohydrate	Flovent Diskus	GlaxoSmithKline	Asthma and COPD
Fluticasone furoate	Lactose monohydrate	Arnuity Ellipta	GlaxoSmithKline	Asthma and COPD
Beclomethasone dipropionate + Formoterol fumarte	Lactose monohydrate, Magnesium stearate	FostairNexthaler	Chiesi	Asthma and COPD
Beclomethasone dipropionate + Formoterol fumarte	Lactose monohydrate	Symbicort Turbohaler	AstraZeneca	Asthma and COPD
Beclomethasone dipropionate + Formoterol fumarte	Lactose monohydrate	DuoRespSpiromax	Teva	Asthma and COPD
Fluticasone furoate + Vilanterol	Lactose monohydrate, Magnesium stearate	Breo Ellipta	GlaxoSMithKline	Asthma and COPD
Fluticasone furoate + Salmeterol	Lactose monohydrate	Advair Diskus	GlaxoSMithKline	Asthma and COPD
Umeclidinium + Vilanterol	Lactose monohydrate, Magnesium stearate	Anoro Ellipta	GlaxoSMithKline	Asthma and COPD
Tobramycin	1,2-distearoyl-sn-glycero-3-phosphocholine, Calcium chloride	TOBI Podhaler	Novartis	Cystic Fibrosis infection
Zanamivir	Lactose	RelenazaDiskhaler	GlaxoSmithKline	Influenza
Insulin Human	Fumaryl diketopiperazine, Polysorbate 80	Afrezza	Sanofi Aventis	Diabetes
Loxapine	N/A	Adasuve	Teva	Schizopherina/Bipolar disorder
Ciprofloxacin	1,2-distearoyl-sn-glycero-3-phosphocholine, Calcium chloride	Ciprofloxacin PulmoSphere	Novartis	Cystic fibrosis

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
