# Peer review of "Dry Powder for Pulmonary Delivery: A Comprehensive Review"

_pharmaceutics, 2020, doi:10.3390/pharmaceutics13010031_

Round 1

Reviewer 1 Report

Dear authors,

It is a well-written review paper containing comprehensive perspectives of DPI. Only a couple of suggestions: 

  1. Figure 1 Particle Clearance section, texts are in vertical direction. Suggest to change and make them more readable
  2. To make it even more comprehensive, suggest to add a section to discuss DPI for different modalities, such as small molecule, proteins (mAbs and others), gene therapy products, vaccines, bacteriophage, etc.
  3. There are some recently emerged particle engineering technologies, please try to include them.
  4. Please consider update your references to as latest as possible for your examples/statements

Author Response

Thanks to reviewer for providing of valuable comments and suggestion towards the improvement of this manuscript. I have tried my best to address all your comments one-to-one. Please, find the response list in upload section.

Thanks 

Reviewer 2 Report

The manuscript submitted intends to provide a comprehensive review of dry powders for pulmonary delivery.  The paper is very comprehensive and well written and organized. The figures and tables are comprehensive and helpful.  Fig. 1 provides nice a comprehensive overview of multiple aspects. Here are some comments and suggestions from a reader’s perspective, and addressing these comments can be helpful to the users.

  1. Page 2 line 45: Please elaborate on “limited tolerance for foreign particles”
  2. Page 2: Line 53-54: Authors reported about the use of some pulmonary compatible materials for coating, it would be helpful to elaborate on such materials (what are they, and how they can help in improving the compatibility?).
  3. Page 3. Section 2: Since each paragraph discusses different aspects, instead of having a big subtitle “Respiratory tract, particles deposition pattern and clearance mechanisms, it would be helpful to breakdown into multiple subsections i) Respiratory tract, ii) particles deposition pattern and iii) clearance mechanisms.
  4. Are there any literature reports providing any relation between
    1. The velocity of the particles vs. inertial impaction
    2. Breath-holding time vs. sedimentation
  5. Page 6, Line 176-177: “Both high and low cohesiveness among particles are problematic for aerosol formation” additional details would be helpful regarding how a low and high cohesiveness affects aerosol formation and performance.
  6. Page 7, Lines 225-228: “Emery, et al., prepared hydroxypropyl methylcellulose (HPMC) and respitose powders with the moisture content of 0%, 2%, 5%, 10% and 0%, 2%, 5% , 8%, respectively, and found that aerosolization of HPMC gradually decreased with increased moisture content, whereas the aerosolization of respitose remained stable [66]”. Why HPMC and respitose materials behaved differently, any rationale/explanation for this behavior?
  7. Minor format (missing space between words)
    1. Line 232: space between depend_on
    2. Line 295: encounter_the
    3. Line 348: is_also
  8. Page 8: Line 248-254: “Matsusyama and Yamamoto have shown that deposition of particles in the airways is also affected by surface charge [76]. Therefore, it is important to optimize the surface charge during the formulation of drugs [77]. The deposition pattern of charged particles in airways has been well explained using computational lung models [78, 79]. Mostly, deposition patterns of dry powder in the airways are governed by inertial impaction, gravitational sedimentation and Brownian diffusion. However, electrostatic charges also contribute to deposition by cohesive attraction and are more relevant for the deposition in lower airways [80].” Does surface charge directly affect the particle deposition (by charge related attraction between the particles and membrane surface) or is it indirectly controlled by aerosolization ability due to charge? Adding details would be helpful.
  9. Page 10: 304-309: It would be helpful to comment on how does the particle size density affected with solvents (if any literature info available)
  10. Page 10, Line 337: “The selection of nozzles is dictated by the desired size of powder production [110].” Adding details if any literature information available on which nozzle results in which size range would be helpful.
  11. Although the authors have provided a nice detailed description under each technique of preparation and made an excellent summary table listing different drug attempted under each dry powder preparation technique, it would be helpful to have a comprehensive summary table for different preparation techniques - comparing principle, size range, charge, pros, and cons for different techniques. This would be an excellent table from a user perspective and that would enhance the paper as well.
  12. Not needed but adding a brief paragraph (a short literature review) on formulation (excipient) strategies that were used to improve the performance of dry powder would complement the review manuscript.

Overall, it is a good and well-attempted comprehensive review manuscript, and if possible addressing the above comments would complement it and can be helpful to the readers.

Author Response

Thanks to reviewer for revising and providing various valuable comments and suggestion towards the betterment of this manuscript. I have tried my best to answer all your comments one-to-one. Please find the response list in upload section. Thanks.

Reviewer 3 Report

The paper under review deals with the formulation of dry powder for pulmonary delivery. The article (review) tackles an important issue in inhalers, aerosolization, deposition, and therefore is suitable for Pharmaceutics. The manuscript includes the information on the physicochemical properties of dry powder, available equipment and manufacturing methods  and presents the research outcomes and their detailed description. A structure of the paper is in accordance with principles of very good scientific review reports. The paper is written in good English. The article contains adequate and appropriately selected 229 literature items.

Comments:

  1. Please check is the literature according to the guide for authors?

In opinion of the reviewer the article can be accepted for publication in Pharmaceutics (ISSN 1999-4923).

Author Response

I am very thankful to reviewer for accepting this manuscript to publish in this Pharmaceutics. 

This whole manuscript has been written based on "author guide".

This manuscript is a resubmission of an earlier submission. The following is a list of the peer review reports and author responses from that submission.

Round 1

Reviewer 1 Report

The paper overall is a nice summary of DPI technology in terms of pulmonary structure, particle properties, and micronization techniques. Even though it is a good information gathering, it is in lack of novelty and logical organization structure. Please see my comments in detail below.

  1. Line 21: I would not use 'sedimentation' since it particularly means deposition by gravity. I would just use deposition
  2. There are many other rapid freeze-drying technologies, such as thin film freezing, droplet freezing, rapid freezing in vials, etc. Please include those and categorize them together with spray freeze drying.
  3. Please rephrase the statements in your pie graph to make them more clear. For example, under 'Surface Charge' section, I assume you are trying to say 'neutral charge leads to better aerosol performance', but it seems very confusing. So does the rest sections.
  4. Some sentences are too long and there are grammar mistakes. Please run a language check by native speakers.
  5. I would split Line 80-111 into two paragraphs: lung structure and deposition patter
  6. Please double check 15 min to 2 h in Line 118. To my knowledge, the clearance of foreign particles can be up to 4 hours.
  7. Mucus is produced by goblet cells in trachea and bronchi (upper than bronchioles and alveoli). Mucus tend to move up and rarely move down to bronchioles and alveoli area. Please add some discussion on how the particle size/shape impact the deposition site, which further leads to increase/decrease in clearance.
  8. Please increase the font in Figure 1. It is not readable in 100% page size. I believe you used Wiebel's tracheobronchial classification. Please revisit the generation numbers of each tracheal region. And please tell your readers what 'Generation' means, what 'Gn' means, etc.
  9. Statement in Line 156-157: span is usually used in geometric particle size measurement. How about GSD in aerodynamic particle size measurement? Also, please add optimum span and GSD range.

  10. Please elaborate more and use more examples for critical attributes, such as particle size (MMAD/Dv50), FPF, GSD/span, ED, etc.
  11. The roughness of particle surface is different from particle shape. A perfect sphere can have rough surface
  12. What is the underneath reason behind the impact of moisture on particle deposition? Electrostatic? Surface morphology? Cohesive force? Please elaborate and support by examples
  13. Please double confirm the statement in Line 229-231. Isn't it the opposite? Rough surface has less contact area.
  14. Please revisit statements in Paragraph 4.1: a). how does milling improve for stability? b). Line 266-267: suitable for meltable materials? c). mechanical stresses is a leading drawback of milling and I did not find a mention of it.
  15. Please reorganize Table 1.
  16. For each particle engineering technology, please add more updated information, such as innovations in equipment and technology, novel excipient/formulation, new application in other modality (for example, gene therapy, biologics, emulsion formulations, etc.)
  17. Please revisit the structure of every paragraph and improve the logic flow.

Author Response

Thanks to reviewer-1 for providing the valuable critiques and suggestions towards the improvement of this manuscript. We apologize for any grammatical mistakes and language flow throughout the manuscript. The whole manuscript is revised thoroughly and is proofread very carefully by a native speaker. Here, we have responded to all critiques one-by-one. We appreciate if you could read the new resubmitted manuscript, which is different from the previous version. Thanks.

Reviewer 2 Report

The manuscript reports a through critical discussion of the particle properties and how they affect inflation performances together with the main preparation techniques.

Albeit overall well written, except for required language minor revision, doubts arise on the impact on readership of such a review.

The literature is full of reviews describing the properties and formulation of dry powders for inhalation. From this review I would rather have expected an insightful discussion on the clinical relevance of such properties according to therapy and disease.

In fact, one important thing the AA have missed in their debate is the fact that the importance of the different properties discussed change according to the therapeutic target in the lungs or systemically.

As it is, this review remains another anonymous report summarizing today well-known issues in pulmonary drug delivery with a relative impact on the readership. Therefore, I suggest to revise the manuscript adding an important section discussing what should be done in order to improve the clinical impact and the building up of proper clinical settings for inhaled products especially in emerging areas, such as infectious diseases or cancer. Moreover, the AA should focus on how relevant are the several different properties described to clinical efficacy and how device developers should cope with it.

A market discussion is also welcome.

Author Response

Thanks to Reviewer-2 for the critiques and for providing valuable suggestion towards the improvement of this manuscript. We apologize for any grammatical mistakes and language flow throughout the manuscript. The whole manuscript is revised thoroughly by a native speaker. Here, we have responded your critique and hope that it can make sense. Basically this manuscript is drafted based on pharmaceutical aspect, so we have not discussed about clinical aspect. But, it is really a nice direction to summarize a new review manuscript considering clinical aspects of DPI. We appreciate if you could read the resubmitted manuscript, which contain lots of changes than the previous version. thanks.  

Reviewer 3 Report

Line 17: the advantages listed mix physiological with user convenience. They should be listed separately for clarity.

Line 18: Improved patient compliance over oral and IV is not supported. Many would disagree since there is data showing poor compliance to inhaled medication regimens is one of the major problems with inhalation therapy.

Line 19: replace “always” with “often”. Replace “its limited” with “restrictions on”

Line 20: Lower capacity compared to what? I would say IV formulations have a lower tolerance to foreign material than pulmonary

Line 22: replace “trajectory” with “tract”. Respiratory trajectory is not term commonly used in the field. Trajectory has a clinical connotation as in “disease trajectory”. It’s use in this manuscript should be replaced with “airways” “respiratory tract” or “pulmonary anatomy”

Line 23: replace “preparation techniques” with “manufacturing methods”

Line 25: Those in the field would not consider spray freeze drying and supercritical fluid as frequently used methods. There are no commercial products that use these methods. I would separate frequently used industrial scale methods (milling, spray drying) and alternative development methods (SFD and supercritical).

Line 26: replace “sedimentation” with “deposition”. Sedimentation is only one of three common method of particle deposition in the airway

Line 27: Replace “comprehends” with “examines”

The manuscript presents a nice overview of the pulmonary anatomy/physiology, important physicochemical properties for aerosolization, and manufacturing techniques. The referenced studies and technical concepts in the paper are generally correct. There is, however, a general bias throughout the manuscript toward pulmonary delivery that is not presented in proper context or well supported. An example form Line 518 states “The pulmonary route is preferred for drug administration as the lungs provide large absorptive surface area and bypass the hepatic portal system”. From this statement I would infer that the author believes all drugs should be administered by inhalation. I realize, of course, this is not the author’s intention, however, sweeping statements and unsupported bias toward pulmonary delivery over other routes should be eliminated from the manuscript. As one who has spent my career working on OIDPs, I agree there are many advantages, but they need to be contextualized and tempered.

This manuscript needs a thorough review for grammar before I would be willing to review again. I stopped providing edits after the abstract as there are too many for me to reasonably cover in this peer review.

It is also obvious that the authors are only focusing on academic studies.  In Table 1, nearly every example is from the academic literature and ignore commercial products. My recommendation would be to focus the paper on only investigational manufacturing techniques and dedicate some discussion to potential for scale up and commercial production.  Unfortunately, I cannot recommend that this manuscript be published in this journal.

Author Response

Thanks to Reviewer-3 for the critiques and providing valuable suggestions towards the improvement of this manuscript. We apologize for any grammatical mistakes and language flow throughout the manuscript. The whole manuscript is revised thoroughly by native speaker. Here, we have responded to all critiques on-by-one. We appreciate if you could read the resubmitted manuscript, which is different from the previous version. Thanks.

Reviewer 4 Report

This is in fact an interesting and useful review. I would support publication, of course. My only suggestion would be that when discussing spray-freeze drying, the authors add discussion of the challenges related to using multi-component systems and the possibilities that such composite "micro-balls" open (see as an example Ogienko, A. G., et al (2017). Large porous particles for respiratory drug delivery. Glycine-based formulations. European Journal of Pharmaceutical Sciences110, 148-156). Maybe the authors can be advised to consider this possibility.  

Author Response

Thanks to Reviewer-4 for providing valuable suggestions. We apologize for any grammatical mistakes and language flow throughout the manuscript. The whole manuscript is revised thoroughly by native speaker. Here, we have responded to your suggestion. We appreciate if you could read the resubmitted manuscript. Thanks

Round 2

Reviewer 1 Report

Dear Authors,

It is a nice summarize of pulmonary delivery related basic knowledge. However, it is in lack of novelty. Most of the content are repeating existing information: more than 3/4 references are older than 5 years and there is no new knowledge in the area introduced.

Thanks for the contribution!

Reviewer 2 Report

My opinion remains the same. As it is, this manuscript is a didactic paper not adding any useful information or contribution to the field. Even conclusions and perspectives do not provide any novelty to current knowledge or critical opinion in the field. Therefore, considering the unsatisfactory response provided, I cannot endorse publication.